# Effect of Lumbar Spine Mobility and Postural Alignment on Menstrual Pain in Young Women

**DOI:** 10.3390/ijerph20156458

**Published:** 2023-07-27

**Authors:** Katarzyna Walicka-Cupryś, Agnieszka Renata Pelc, Mirosław Pasierb, Sylwia Płonka, Agnieszka Pal, Nikola Rosowska

**Affiliations:** 1Faculty of Health Sciences, College of Medical Sciences, University of Rzeszów, Rejtana 16c Street, 35-959 Rzeszów, Poland; 2Student Scientific Circle—Fascination with Body Posture “Habitus”, University of Rzeszów, Rejtana 16c Street, 35-959 Rzeszów, Poland

**Keywords:** menstruation, pain, spinal curvature, lumbar lordosis, inclinometer, physiotherapy, osteopathy, gynaecology, forward head, dowager’s hump, smartphone hump

## Abstract

(1) Background: Studies indicate that 45 to 95% of women suffer from painful periods. Despite frequent incidence, the problem is still underestimated. Menstruation-related ailments often correlate with high absenteeism at school and work, as well as limitation of social and scientific activity. The aim of this study was to assess the relationship between the size of the curvature, the mobility of the spine in the sagittal plane, and menstrual pain in young women. (2) Methods: The analysis included 224 women aged 18–26, mean of 21.56 ± 1.95, studying at the College of Medical Sciences of the University of Rzeszów. For this purpose, the measurement of the anterior–posterior curvature of the spine with a gravitational inclinometer, the Schober test and the authors’ questionnaire related to menstruation were used. (3) Results: The data analysis demonstrated a significant difference between the group with menstrual pain (MP) and the control group without pain (CG), where the angle of the lumbar sacral transition (*p* = 0.034), thoracolumbar transition (*p* = 0.044) and lumbar lordosis (*p* = 0.006) were statistically significantly lower in MP compared to CG. There were no significant differences in the measurement of the so-called smartphone hump and between the mobility of the lumbar spine and menstrual pain in any of the groups. (4) Conclusions: Women with menstrual pain are characterized by reduced lumbar lordosis and thoracic kyphosis, while sagittal mobility of the spine does not affect pain.

## 1. Introduction

The menarche is associated with many fears in young women, and knowledge about the physiological processes is limited [1]. Menstruation is a natural, biological process that marks a girl’s transition into adulthood. Menstruation is the first phase of the menstrual cycle, during which the endometrial epithelium is shed under the influence of sex hormones [2]. Studies indicate that 45 to 95% of women have painful periods. Despite such a frequent incidence, the problem is still underestimated [3,4]. Pain most often affects women under 25 and decreases with age. Dysmenorrhea is characterized by cramping in the lower abdomen that starts at the beginning of the cycle and lasts 8 to 72 h. Some women may also experience pain radiating to the back of the legs or the lower back [5]. Pain is accompanied by nausea, vomiting, diarrhoea, headaches, muscle cramps, breast tenderness, low back pain, fatigue, increased appetite and even sleep disorders and mood changes [6,7,8]. Studies show that just before menstruation, the uterus is enlarged [9]. This can cause pressure on the surrounding tissues, including nerve endings, contributing to the onset of pain. This situation may be intensified by the change in the inclination of the sacrum and the curvature of the spine in the sagittal plane, further reducing the space for the swollen uterus, all the more intensifying the compression and possible pain.

Apart from non-pharmacological methods of treatment, painkillers play an important role. However, women who use pharmacotherapy to treat pain must be aware of the significant risk of side effects [10,11]. There are two types of dysmenorrhea: primary dysmenorrhea, which occurs without a diagnosed pelvic pathology, and secondary dysmenorrhea caused by a specific gynecological disorder, such as endometriosis [12,13], endometrial polyps, or the use of intrauterine contraceptives [7].

Numerous causes have been found in the literature that affect menstrual pain, which leads to a reduced quality of life, absenteeism and an increased risk of depression and anxiety [14]. According to the data, painful menstruation is the main cause of recurrent absences from school or work [6].

An analysis of the available studies indicates that the incorrect position of the spine and pelvis as well as changes in the structure of the abdominal muscles are the source of painful menstruation. According to Karakus et al., the strength, thickness, position and mobility of the muscles of the lumbar–pelvic complex have a significant impact on the level of pain in women during menstruation [15]. Meanwhile, Moon-jeong Kim et al. analyzed the negative correlation between the position of the pelvis and the thickness of the abdominal muscles and pain [16]. Another study by Olados et al. shows that abdominal muscle activity contributes to painful menstruation [11].

There are studies on the correlation between painful menstruation and the musculoskeletal system; however, no analysis related to the assessment of spinal curvature and pain during menstruation is available [15]. It is assumed that musculoskeletal pathology will cause changes in the structure and function of the body, including a change in the position of the uterus, which leads to an increase in dysmenorrhea [16]. Therefore, the aim of this study was to assess the relationship between the curvature and mobility of the spine in the sagittal plane on menstrual pain in young women.

## 2. Materials and Methods

### 2.1. Study Participants

The analysis included 224 women aged 18–26, with a mean age of 21.56 ±1.95, studying at the College of Medical Sciences of the University of Rzeszów, with a mean body weight of 61.6 kg (±9.4), mean body height of 166.1 cm (±5.93) and BMI of 22.3 (±2.9) kg/m^2^. Based on the questionnaire, a study group with menstrual pain (MP) (*n* = 146; 65.18%) and a control group without menstrual pain (CG) (n = 78; 34.82%) were selected. The study group consisted of women with menstrual pain ≥ 4, and the control group, women with pain < 4 on the Visual Analogue Scale (VAS scale) [17], who met the inclusion criteria. The mean body height in CG was 165.9 (±6.3) cm and in MP, 166.2 (±5.8) cm, and the mean body weight in CG was 61.7 (±9.0) kg and in MP, 61.5 (±9.7) kg. The mean BMI in CG was 22.4 (±2.7) kg/m^2^ while in MP it amounted to 22.2 (±3.0) kg/m^2^.

The inclusion criteria for the study group included: women aged 18–26, studying at the College of Medical Sciences of the University of Rzeszów, menstrual pain ≥ 4 on the VAS scale [17].

The exclusion criteria from both groups were: women not studying at the College of Medical Sciences of the University of Rzeszów, diagnosed diseases of the reproductive system and/or within the spine, and pregnant women and those in the postpartum period.

The criteria for the inclusion of the control group were: women aged 18–26, studying at the College of Medical Sciences of the University of Rzeszów, menstrual pain < 4 on the VAS scale and no diagnosed diseases of the reproductive system or within the spine. The exact flow of the respondents is presented on Figure 1.

### 2.2. Study Qualification

The study was divided into three stages. In the first stage, an invitation to participate in a study was distributed among the women studying at the College of Medical Sciences of the University of Rzeszów (*n* = 325). A total of 224 women aged 18–26 participated in the second stage. A given stage consisted of anthropometric examination: height and weight, tests of the curvature of the spine with an inclinometer and the mobility of the lumbar segment with a Schober test. In the third stage, the examined women completed an anonymous questionnaire on menstruation.

The minimum study group was calculated on the basis of the sample size calculator [18,19]. Data for estimating the smallest sample size were calculated on the basis of the number of 2838 women studying at the College of Medical Sciences at the University of Rzeszów. The statistical power of our study was 0.88, the recommended minimum power was 0.8, and the required number of people in the group was 223.

### 2.3. Methods

#### 2.3.1. Anthropometric Measurements

All the measurements were performed on the same day, starting with the anthropometric measurements. Body height was measured with a Seca 213 mobile stadiometer, with an accuracy of 0.1 cm. Body mass was measured using the electronic scale OMRON BF 500, with an accuracy of 0.1 kg. The measurements were performed under standard conditions; women in underwear and barefoot stood upright without bending the knees.

#### 2.3.2. Assessment of Spinal Curvature

The gravitational inclinometer in a smartphone was used to assess the parameters of anterior–posterior curvature of the spine, according to the methodology of Walicka-Cupryś K. et al. [20]. The assessment was conducted in a standing position, with straight knees, looking at a point in front of the subject at a distance of 1.5 m. The following instructions were given: “stand comfortably”, “do not bend your knees” and “look straight”. The subjects were not told to straighten up [20,21,22].

The study covered the following parameters:ALPHA 1 angle—inclination of the sacrum (upper beam of the inclinometer in the middle of the intervertebral space on the line connecting the posterior superior iliac spines).ALPHA 2 angle—sacrolumbar junction S/L (centre of the inclinometer on the line connecting the posterior superior iliac spines).BETA angle—Th12-L1 thoracolumbar junction (centre of the inclinometer at the thoracolumbar junction).GAMMA angle—C7-Th1 (upper beam of the inclinometer on C7).DELTA angle—Th3-Th4 (upper beam of the inclinometer at the height of the anglus superior of the scapula). The location of the parameters is presented in Figure 2.

Changes in the curvature of the spine were assessed on the basis of the general guidelines for inclinometers according to Saunders, where the following values were accepted as correct:Lumbosacral angle (ALPHA 2 angle)—15–30°;Lumbar lordosis angle—LLA (ALPHA 2 angle + BETA angle)—30–40°;Thoracic kyphosis angle –TKA (BETA angle + DELTA angle)—30–40° [23];Middle thoracic kyphosis angle—MTKA (BETA angle + GAMMA angle)—30–40° [20,22];A so-called smartphone hump—SH (also known as a dowager’s hump [24]) was considered when (GAMMA angle—DELTA angle) ≥ 21° [20].

#### 2.3.3. Assessment of Spinal Mobility

The non-invasive Schober test was used to assess the mobility of the lumbar spine. Two points on the patient’s skin were marked: the first at the level of the L5 vertebra, and the second 10 cm in the cranial direction. Then, a centimetre tape was applied to both points. The patient made the maximum flexion of the trunk with straightened knee joints (the norm is a difference of 5 cm), then the maximum extension of the trunk was performed (the norm is a difference of 1–2 cm) [15,25].

#### 2.3.4. Questionnaire

The authors’ questionnaire contained 31 items. The basic questions concerned data such as age, place of residence, education and possible incidence of diseases. The main part of the questionnaire included questions about the age of menarche, average cycle length, intensity of menstrual bleeding (i.e., the amount of blood lost during menstruation), symptoms accompanying menstruation and determination of the intensity of pain experienced by the subjects. Pain intensity was assessed using the Visual Analogue Scale (VAS), which consisted of a horizontal line 10 cm long with extreme descriptors, with the left side signifying no pain and the right side signifying the worst imaginable pain. The subjects indicated their individual pain magnitude by marking a spot on the line. A ruler was used to quantify the measurement results on a scale of 0 to 100 mm. The questionnaire also contained questions regarding information on methods of pain relief used by the respondents [26].

### 2.4. Statistical Analysis

The statistical analysis of the collected material was carried out in the Statistica 13.3 package. The normality of distributions was verified with the Shapiro–Wilk W test, and the equality of variances was assessed with Levene’s test.

Due to the lack of equality of the distribution and the different numbers in the groups, a non-parametric test for independent samples, the Mann–Whitney U test, was used to analyse the significance of differences between the groups in the analysed parameters. Pearson’s Chi-squared test was used to assess the frequency of a given change in the curvature of the spine between the groups. The Spearman’s rank correlation test was used to assess the relationship between the analyzed parameters in the spine and selected quantitative data related to menstruation. Statistical significance was assumed at *p* < 0.05.

## 3. Results

A statistically significant difference between the control group and the study group was observed in terms of symptoms accompanying menstruation, where worse parameters were recorded in the group with menstrual pain and were: lack of concentration (*p* < 0.001), limitation of physical activity (*p* < 0.001), adverse effect on the results in study/work (*p* < 0.001) and worse feeling during menstruation (*p* < 0.001). However, elevated body temperature (*p* = 0.13) and absence from work/university (*p* = 0.088) were not statistically significant (Table 1).

The data analysis showed a significant difference between the study group and the control group, where the ALPHA 2 angle and the thoracolumbar transition BETA was statistically significantly lower in the study group (17.1°) compared to the control group (18.9°) respectively (*p*= 0.034, *p* = 0.044). The analysis of the size of the lumbar lordosis angle showed significantly lower parameters in the study group (28.5°) than in the control group (32.2°), (*p* = 0.006). Similarly, the size of the middle thoracic kyphosis parameters was statistically significantly different (*p* = 0.037) between the groups: the study group was characterized by significantly smaller parameters (22.3°) than the control group (24.9°). The angle of inclination of the thoracic kyphosis angle was close to statistical significance at *p* = 0.063, but no significant differences were found in the measurement of the so-called smartphone hump. There were no significant statistical differences in the mobility of the lumbar spine measured by the Schober test and menstrual pain in any of the study groups (Table 2).

The data analysis showed a significant difference in the mean length of bleeding in the study group compared to the control group (*p* = 0.017). Statistically significant differences in the pain before menstruation between the study group and the control group were found (*p* = <0.001). The difference in menstrual cycle length in days was close to statistical significance between the groups (*p* = 0.057) (Table 3).

The analysis of spinal curvature norms between the study group and the control group did not reveal any statistically significant differences in the examined parameters. Both in the study group and in the control group, the parameters in most cases did not exceed the norm. Particularly noteworthy is the MTKA parameter (*p* = 0.067), as a significant proportion of the subjects showed results below the norm (Table 4).

Interpretation of the data on methods of menstrual pain relief showed that almost one-third of the surveyed female students reported the use of analgesics during each menstruation, and another 41.1% of female students take them, but not during each menstruation. Only 23.3% of the respondents in the study group consulted a doctor about painful menstruation.

There were no significant correlations between the parameters of spinal curvature and its mobility and pain. In the study group, significant, positive correlations were found between the size of the thoracolumbar junction and the MTKA and TKA with the intensity of bleeding, respectively, R = 0.25, R = 0.17, and R = 0.21, and menstruation-related nuisance, respectively, R = 0.25, R = 0.17, and R = 0.19, i.e., the greater the kyphosis, the more severe the pain and bleeding. Such correlations were not found in the control group, with the exception of the DELTA parameter and the regularity of bleeding, i.e., the higher the DELTA parameter, the more regular the bleeding, R = 0.22 (Table 5).

## 4. Discussion

In the literature on the subject, a limited number of studies is available on the connection between the angle of inclination of the spinal curvature in the sagittal plane and painful menstruation [15,16]. Considering that the state of knowledge is insufficient, research in this direction is highly recommended. Most authors associate menstrual pain with lifestyle, i.e., diet [27], BMI [28], physical activity [29] or stress [30]. The results of this study confirm that the problem of menstrual pain concerns a large number of women and is related to the size of the spinal curvature and not their mobility.

Numerous sources state [27,28,29,30,31] that primary dysmenorrhea is a very common phenomenon among adolescent girls of junior high school age and among female students. Menstruation-related complaints often correlate with high absenteeism at school and work, as well as limitation of social and scientific activity [32,33,34,35,36]. Our study included women in a similar age range.

In their work, Pembe et al. report that as many as 23.6% of young girls left school due to dysmenorrhea [37]. Yücel et al. studied menstruating girls aged 9–18. Their work shows that 15.9% of all respondents leave school because of menstruation, and in the group that experienced pain during menstruation, 18% of respondents were absent [35]. According to other authors, pain makes it difficult to perform daily activities in 41.7% of respondents, and 25.1% of women reported absence from school or university [38]. Oladosu et al. also observed a greater absence from work or school due to menstrual pain, amounting to 2.2 ± 0.4 days [11], while our study showed an absence of 1.2 days.

Teul et al. showed that back pain is associated with painful menstruation, as it was significantly less frequent in the group with painless menstruation [39]. Similar conclusions were drawn by Knapik et al., who showed the presence of menstrual-related back pain in 8.6% of all menstruating girls covered by the study [40], which is not consistent with the results of this study.

Karakus et al. found a relationship between the mobility of the pelvis in the sagittal plane and the anterior lumbar angle with primary dysmenorrhea compared to asymptomatic women [15]. The angle of lordosis was much larger (37.4) than in our study (28.5). Kim et al. showed that the lordotic angle was greater in the group of women with dysmenorrhea [16]. The average angle of inclination of the lumbar lordosis in our research was lower than the recommended normative angle (30°–40°) [23], which, according to the literature, may contribute to the occurrence of pain in the lower spine [41]. At the same time, pain in the case of shallowing of the lumbar lordosis may probably be caused by reduced space for internal organs and a swollen uterus. Our assumptions may be confirmed by the fact that, as reported by Molins-Cubero et al., after repositioning of the pelvis/pelvic manipulation in women with primary pain, the subjective sensation of menstrual pain was reduced [42].

Rakhshaee et al. aimed to assess the impact of practicing yoga on the reduction of menstrual pain in women aged 18–22. The women in the study group were tasked with performing specific yoga asanas on a daily basis. The pain level was measured during three consecutive menstrual cycles. An analysis of the results showed that the level and duration of women’s menstrual pain significantly decreased after the exercise [43]. The same conclusions were reached by Yonglitthipagon et al. and Yang et al., who conducted similar studies [44,45]. Yoga has a positive effect on the human body [46]. It improves the flexibility of the spine, strengthens the back muscles, coordinates movement and breathing and relieves muscle stiffness [43]. This study did not show statistically significant differences in the mobility of the lumbar spine, measured with the Schober test, between the study groups differing in menstrual pain. However, various exercises can affect the very shape of the lumbar lordosis and normalize its parameters, which would explain the reduction in or elimination of pain, but this requires further research.

In the treatment of menstrual pain, pharmacotherapy is commonly used, in particular non-steroidal anti-inflammatory drugs [10]. Yücel et al. reported that 34% of the young girls surveyed took analgesics during menstruation [35]. As many as 92% of female students investigated by Torkan et al. took at least one analgesic drug during the first three days of menstruation [36].

Severe pain during menstruation may also be a symptom of diseases of the female reproductive system [43]. Endometriosis is a complex syndrome characterized by a chronic inflammatory process conditioned by the activity of estrogens, which mainly affects the pelvic tissues, including the ovaries [47]. Most women are asymptomatic, but some women have severe dysmenorrhea, non-cyclical chronic pelvic pain, uterine bleeding, infertility, dyspareunia, painful menstrual defecation and urinary and gastrointestinal symptoms [48]. Sara Clemenza et al. presented endometriosis as the most common cause of secondary dysmenorrhea. Research by Federic Facchin et al. showed that patients with endometriosis and pelvic pain are characterized by a worse quality of life and mental health compared to patients with asymptomatic endometriosis and healthy people [31].

It is estimated that polycystic ovary syndrome occurs in 5 to 20% of women of reproductive age [49,50]. The incidence of PCOS in our studies was slightly lower. Studies by Jeong et al. confirm that the disease can significantly increase the level of pain experienced during menstruation and also cause increased bleeding [51], which is also confirmed by this study.

Research by Ramos-Pichardo et al. conducted on Spanish students showed that most of them consider menstrual pain something normal and natural. Only 44.19% of the young women surveyed had previously consulted a doctor due to painful menstruation. Others treated themselves with analgesics or non-pharmacological methods of pain therapy [52]. Our study presents similar statistics. Far too few young women consult a doctor in the event of painful menstruation. There is a suspicion that some women have health problems/diseases of the reproductive system that they are not aware of. For this reason, special attention should be paid to broader education and health promotion for women of all ages.

## 5. Limitations

The present study has some limitations. We recognize that the age at menarche may have played a role in responses on menstrual pain. Each of the patients also had different conditions of private life, knowledge about the state of physical health and emotional approach. We think it would be interesting to study a group of women who experienced menarche at the same age and were of similar height and weight while remaining within the exclusion criteria for the study. It would give a picture less distorted by these factors. Scientific studies show that there is a visible relationship between the angle of inclination of the lumbar spine and menstrual pain. However, it would be worth conducting further research between groups of women without menstrual pain and women with pain who had their first menstrual period at the same age, which may show differences. Additional studies will be needed to investigate the correlation between the size of the curvature of the spine and the location of the internal organs.

## 6. Conclusions

Our results suggest that a shallower lumbar lordosis and thoracic kyphosis is observed in women with menstrual pain. In other words, a flat back increases the likelihood of perimenstrual pain among women. This study found no correlation between the smartphone hump angle and menstrual pain. The mobility of the spine in the sagittal plane measured with the Schober test was also not significant in relation to pain. These results may be useful in the diagnosis and treatment of women with primary and secondary menstrual pain.

## Figures and Tables

**Figure 1 ijerph-20-06458-f001:**
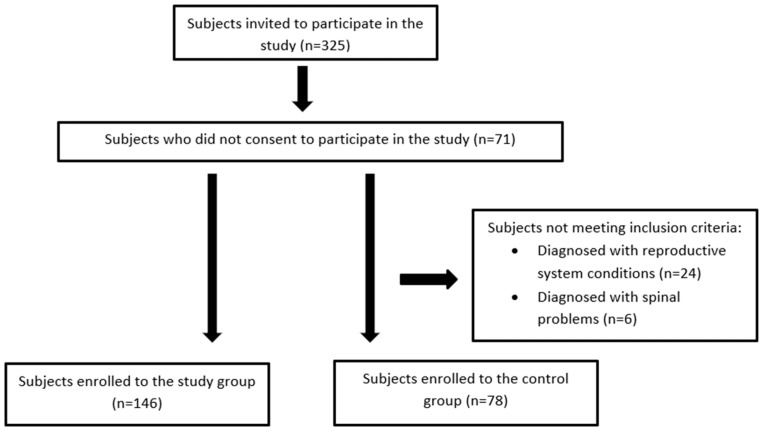
Flow chart of the study participants.

**Figure 2 ijerph-20-06458-f002:**
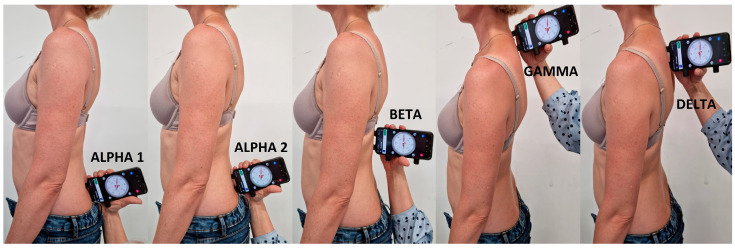
Location of the parameters measured in the study.

**Table 1 ijerph-20-06458-t001:** Assessment of symptoms accompanying menstruation between the study group and the control group.

	Control Group		Study Group			
Outcome Variable	x	S	x	S	U	*p*
Absence from work/university	0.03	0.16	0.16	0.37	−3.08	0.088
Lack of concentration	0.17	0.38	0.60	0.49	−6.23	**<0.001**
Limited physical activity	0.38	0.49	0.79	0.41	−6.00	**<0.001**
Adverse impact on academic/work performance	0.13	0.34	0.45	0.50	−4.78	**<0.001**
Increased body temperature	0.18	0.39	0.30	0.46	−1.98	0.133
Worse feeling compared to the days before menstruation	0.62	0.49	0.91	0.29	−5.34	**<0.001**

x—arithmetic mean; S—sample standard deviation; U—the test statistic for the Mann–Whitney U Test; *p*—probability value; bold—statistical significance at *p* < 0.05.

**Table 2 ijerph-20-06458-t002:** Assessment of spinal curvature parameters between the study group and the control group.

	Control Group			Study Group				
Outcome Variable	x	Me	S	x	Me	S	U	*p*
ALPHA 1	21.4	22.0	5.3	20.4	20.0	5.4	1.62	0.106
ALPHA 2	18.9	18.0	6.3	17.1	17.0	6.9	2.13	**0.034**
BETA	13.3	14.0	5.6	11.5	11.5	6.9	2.02	**0.044**
GAMMA	26.1	26.0	6.4	26.1	24.0	14.1	1.29	0.197
DELTA	11.6	11.0	5.1	10.8	10.0	5.8	1.33	0.184
LLA	32.2	32.5	9.4	28.5	29.0	10.3	2.74	**0.006**
MTKA	24.9	25.0	7.7	22.3	22.5	9.7	2.08	**0.037**
TKA	39.4	39.5	9.6	37.6	35.5	16.8	1.86	0.063
SH	14.4	14.0	6.5	15.3	13.0	14.1	0.79	0.431
Schober Test—flexion	4.1	4.0	1.2	4.2	4.0	1.7	0.00	0.999
Schober Test—extension	1.9	1.5	1.0	1.9	2.0	0.9	−0.70	0.497

ALPHA 1inclination of the sacral bone; ALPHA 2—inclination of sacrolumbar junction S/L; BETA—inclination of the Th12-L1; GAMMA—inclination of the C7-Th1; DELTA—inclination of the Th3-Th4; LLA—lumbar lordosis angle; MTKM—middle thoracic kyphosis angle; TKA—thoracic kyphosis angle; SH—smartphone hump angle; x—arithmetic mean; Me—median; S—sample standard deviation; U—test statistic for the Mann–Whitney U test; *p*—probability value; bold—significance at *p* < 0.05.

**Table 3 ijerph-20-06458-t003:** Assessment of the mean length of bleeding between the study group and the control group.

	Control Group			Study Group				
Outcome Variable	x	Me	S	x	Me	S	U	*p*
Mean length of bleeding (days)	5.2	5.0	1.0	5.7	6.0	1.0	−2.52	**0.017**
Pain before menstruation (days)	0.8	0.0	1.4	2.0	1.0	2.2	−5.26	**<0.001**
Duration of menstrual cycle (days)	28.6	28.0	1.9	29.2	29.0	2.5	−1.95	0.057

x—sample mean; Me—median; s—sample standard deviation; U—the test statistic for the Mann–Whitney U Test; *p*—probability value; bold—statistical significance at *p* < 0.05.

**Table 4 ijerph-20-06458-t004:** Assessment of spinal curvature between the study group and the control group according to the normative.

	Control Group						Study Group						*p*
Outcome Variable	Below Norm		Norm		Above Norm		Below Norm		Norm		Above Norm		
	n	%	n	%	n	%	N	%	n	%	n	%	
ALPHA 1	9	11.5	66	84.6	3	3.9	23	15.8	117	80.1	6	4.1	0.886
LLA	29	37.2	34	43.6	15	19.2	79	54.1	54	37.0	13	8.9	0.261
MTKA	55	70.5	22	28.2	1	1.3	114	78.1	28	19.2	4	2.7	0.067
TKA	11	14.2	29	37.2	38	48.7	32	21.9	59	40.4	55	37.7	0.822
SH	-	-	69	88.5	9	11.5	-	-	125	85.6	21	14.4	0.699

ALPHA 1—inclination of the sacral bone; LLA—lumbar lordosis angle; MTKM—middle thoracic kyphosis angle; TKA—thoracic kyphosis angle; SH—smartphone hump angle; n—number of observations; *p*—probability value of Chi-squared test; bold—significance at *p* < 0.05.

**Table 5 ijerph-20-06458-t005:** Assessment of the relationship between the parameters of the curvature of the spine and the sensations associated with menstruation.

Spearman’s Rank-Order Correlation				
	Control Group		Study Group	
A Pair of Variables	R Spearman	*p*	R Spearman	*p*
ALPHA 1 and medium pain	−0.15	0.189	0.07	0.369
ALPHA 2 and medium pain	−0.13	0.239	−0.11	0.206
BETA and medium pain	−0.06	0.620	0.11	0.204
DELTA and medium pain	0.06	0.604	−0.06	0.494
GAMMA and medium pain	0.17	0.127	−0.02	0.827
LLA and medium pain	−0.11	0.336	0.03	0.687
MTKA and medium pain	0.05	0.682	0.05	0.516
TKA and medium pain	0.07	0.548	0.08	0.359
SH and medium pain	0.17	0.142	0.03	0.718
Schober Test—flexion and medium pain	0.11	0.328	0.04	0.659
Schober Test—extension and medium pain	0.17	0.145	−0.03	0.762
ALPHA 1 and bleeding intensity	0.08	0.472	0.04	0.651
ALPHA 2 and bleeding intensity	0.01	0.930	−0.05	0.589
BETA and bleeding intensity	0.12	0.284	**0.25**	**0.002**
DELTA and bleeding intensity	0.10	0.401	−0.00	0.974
GAMMA and bleeding intensity	0.14	0.237	0.05	0.556
LLA and bleeding intensity	0.07	0.517	0.14	0.102
MTKA and bleeding intensity	0.16	0.153	**0.17**	**0.046**
TKA and bleeding intensity	0.14	0.226	**0.21**	**0.013**
SH and bleeding intensity	0.06	0.582	0.08	0.311
Schober Test—flexion and bleeding intensity	0.08	0.470	−0.03	0.691
Schober Test—extension and bleeding intensity	0.10	0.394	0.05	0.574
ALPHA 1 and periods regularity	0.05	0.667	−0.06	0.500
ALPHA 2 and periods regularity	0.16	0.173	0.05	0.568
BETA and periods regularity	−0.02	0.836	−0.14	0.090
DELTA and periods regularity	**0.22**	**0.049**	0.01	0.933
GAMMA and periods regularity	0.14	0.236	0.14	0.085
LLA and periods regularity	0.09	0.447	−0.04	0.602
MTKA and periods regularity	0.10	0.392	−0.07	0.410
TKA and periods regularity	0.02	0.880	0.04	0.663
SH and periods regularity	−0.08	0.484	0.12	0.140
Schober Test—flexion and periods regularity	−0.01	0.897	0.04	0.631
Schober Test—extension and periods regularity	0.04	0.731	0.00	0.978
ALPHA 1 and menstruation related nuisance	−0.03	0.770	0.11	0.176
ALPHA 2 and menstruation related nuisance	−0.17	0.138	−0.03	0.741
BETA and menstruation related nuisance	0.11	0.345	**0.25**	**0.002**
DELTA and menstruation related nuisance	−0.08	0.487	0.00	0.978
GAMMA and menstruation related nuisance	−0.09	0.411	0.04	0.630
LLA and menstruation related nuisance	−0.05	0.635	0.14	0.090
MTKA and menstruation related nuisance	0.06	0.574	**0.17**	**0.040**
TKA and menstruation related nuisance	−0.00	0.975	**0.19**	**0.025**
SH and menstruation related nuisance	0.06	0.621	0.06	0.507
Schober Test—flexion and menstruation related nuisance	−0.06	0.601	−0.06	0.470
Schober Test—extension and menstruation related nuisance	0.03	0.765	−0.08	0.338

ALPHA 1—inclination of the sacral bone; ALPHA 2—inclination of the sacrolumbar junction S/L; BETA—inclination of the Th12-L1; GAMMA—inclination of the C7-Th1; DELTA—inclination of the Th3-Th4; LLA—lumbar lordosis angle; MTKM—middle thoracic kyphosis angle; TKA—thoracic kyphosis angle; SH—smartphone hump angle; p—probability value; bold—significance at *p* < 0.05.

## Data Availability

Not applicable.

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
