# Peer review of "Effect of Lumbar Spine Mobility and Postural Alignment on Menstrual Pain in Young Women"

_ijerph, 2023, doi:10.3390/ijerph20156458_

Round 1
Reviewer 1 Report (Previous Reviewer 1)
I appreciate the efforts of the Authors in improving their manuscript and taking into account the suggestions of the reviewers.
The new version of the manuscript is prepared with due care and in accordance with the editorial rules required by IJERPH. However, I suggest some point that need correction:
- Why did the authors use the abbreviation "the VAS scale" in line 83 and explain it in line 89? The explanation of an abbreviation should be in the place of its first use in the text.
- The content in lines 158-163 needs improvement. Authors should describe the VAS pain scale they used in the study, and not quote a description of the version usually used.
It should be written, for example, like this: "The main part of the questionnaire included questions about the age of menarche, average cycle length, intensity of menstrual bleeding and symptoms accompanying menstruation, and determination of the intensity of pain experienced by the subjects. Pain intensity was assessed using the Visual Analogue Scale (VAS), which consisted of a horizontal line 10 cm long with extreme descriptors, where the left side signifying no pain and the right side signifying the worst imaginable pain. Subjects indicated their individual pain magnitude by marking a spot on the line. A ruler was used to quantify the measurement results on a scale of 0 to 100 mm."
Author Response
Agnieszka Pelc
Department of Physiotherapy, Institute of Health Sciences, Medical College of Rzeszow University
International Journal of Environmental Research and Public Health
Editor: Ms. Julie Liu
Dear Editor-in-Chef,
Please find enclosed the manuscript Effect of lumbar spine mobility and postural alignment on menstrual pain in young women ID: ijerph- 2392669, which we would like to submit for publication as a full research paper in: The Role of Physiotherapy and Osteopathy in Gynecology and Obstetrics Issues after changes according to the remarks of reviewers. We thank the reviewers for all their comments. Answers are summarized in the table below. According to the comments, we state that a native speaker has verified the text.
|
No. |
CONTENT |
PAGE (LINES) |
CHANGE MODE |
|
Review 1 |
|||
|
1. |
I appreciate the efforts of the Authors in improving their manuscript and taking into account the suggestions of the reviewers. |
n/a |
Thank you for your opinion. |
|
2. |
The new version of the manuscript is prepared with due care and in accordance with the editorial rules required by IJERPH. However, I suggest some point that need correction: Why did the authors use the abbreviation "the VAS scale" in line 83 and explain it in line 89? The explanation of an abbreviation should be in the place of its first use in the text. |
2 (83-89) |
Thank you for your guidance. We have applied the corrections as suggested. |
|
3. |
The content in lines 158-163 needs improvement. Authors should describe the VAS pain scale they used in the study, and not quote a description of the version usually used. It should be written, for example, like this: "The main part of the questionnaire included questions about the age of menarche, average cycle length, intensity of menstrual bleeding and symptoms accompanying menstruation, and determination of the intensity of pain experienced by the subjects. Pain intensity was assessed using the Visual Analogue Scale (VAS), which consisted of a horizontal line 10 cm long with extreme descriptors, where the left side signifying no pain and the right side signifying the worst imaginable pain. Subjects indicated their individual pain magnitude by marking a spot on the line. A ruler was used to quantify the measurement results on a scale of 0 to 100 mm." |
4 (158-163) |
Thank you for your guidance. We have applied the corrections as suggested. |

Reviewer 2 Report (New Reviewer)
The text is well written. The entire study is clearly described and prompts the reader to finish reading.
The theme is original and has not been sufficiently developed in the past.
The introduction is sufficient for the subject at hand. The description of inclusion and exclusion criteria is clear. Also, in detail the demographic data of the sample. The measurements and the method followed are clear.
The necessary normality checks have been made. The results and discussion are consistent with how the study was conducted.
If a clarification had to be made it would be in relation to the intensity of the bleeding. I would ask the authors to add it.
Author Response
Agnieszka Pelc
Department of Physiotherapy, Institute of Health Sciences, Medical College of Rzeszow University
International Journal of Environmental Research and Public Health
Editor: Ms. Julie Liu
Dear Editor-in-Chef,
Please find enclosed the manuscript Effect of lumbar spine mobility and postural alignment on menstrual pain in young women ID: ijerph- 2392669, which we would like to submit for publication as a full research paper in: The Role of Physiotherapy and Osteopathy in Gynecology and Obstetrics Issues after changes according to the remarks of reviewers. We thank the reviewers for all their comments. Answers are summarized in the table below. According to the comments, we state that a native speaker has verified the text.
|
No. |
CONTENT |
PAGE |
CHANGE MODE |
|
Review 2 |
|||
|
1. |
The text is well written. The entire study is clearly described and prompts the reader to finish reading. |
n/a |
Thank you for your opinion. |
|
2. |
The theme is original and has not been sufficiently developed in the past. |
n/a |
Thank you for your opinion. |
|
3. |
The necessary normality checks have been made. The results and discussion are consistent with how the study was conducted. |
n/a |
Thank you for your opinion. |
|
4. |
If a clarification had to be made it would be in relation to the intensity of the bleeding. I would ask the authors to add it. |
4 |
Thank you for your guidance. We have applied the corrections as suggested. We added information on the intensity of bleeding. |

Reviewer 3 Report (New Reviewer)
This study investigated the relationship between the spinal alignment parameters and menstrual pain in young women. The authors found that the angle of the lumbar sacral transition, thoracolumbar transition, lumbar lordosis was significantly lower in the menstrual pain group than in the control group.
The main problem in this study that I concerned is that the authors measured the spinal parameters by using a gravitational inclinometer in a smartphone. This is an indirect method and the results may be influenced by geometry of vertebrae, muscle, and subcutaneous fat tissue. Since the measured topographic points were on the surface of the body, manual palpation of these points may also introduce bias. Traditionally, the spinal parameters should be measured based on the radiography or other medical images. Therefore, the authors should justify the reasonability of the methodology used in this study.
Another problem is that this study investigated the relationship of spinal alignment and menstrual pain, but not the causal effect of these two factors. Therefore, some representation (e.g., effect on...) in the paper should be modified.
Several minor issues include:
Line 91, the reverse of the inclusion criteria should not occur in the exclusion criteria.
Line 121, the authors may add a graph discribing the location of the parameters measured in the study. This could let readers understand the paper more easily.
Some representation can be improved.
For example, in "size of curvatures" (line 74), "size of" can be removed.
"Mean body height in CG 165.9 (±6.3) cm" (line 84) lacks predicate.
"Mann-Whitney U test" (line 172). This sentence seems to be imcomplete.
"The results of our study give similar" (line 307). This sentence seems imcomplete.
Therefore, the paper should be polished by a native speaker.
Author Response
Agnieszka Pelc
Department of Physiotherapy, Institute of Health Sciences, Medical College of Rzeszow University
International Journal of Environmental Research and Public Health
Editor: Ms. Julie Liu
Dear Editor-in-Chef,
Please find enclosed the manuscript Effect of lumbar spine mobility and postural alignment on menstrual pain in young women ID: ijerph- 2392669, which we would like to submit for publication as a full research paper in: The Role of Physiotherapy and Osteopathy in Gynecology and Obstetrics Issues after changes according to the remarks of reviewers. We thank the reviewers for all their comments. Answers are summarized in the table below. According to the comments, we state that a native speaker has verified the text.
|
No. |
CONTENT |
PAGE |
CHANGE MODE |
|
Review 3 |
|||
|
1. |
This study investigated the relationship between the spinal alignment parameters and menstrual pain in young women. The authors found that the angle of the lumbar sacral transition, thoracolumbar transition, lumbar lordosis was significantly lower in the menstrual pain group than in the control group. |
n/a |
Thank you, your observation is correct. Parameters in the group with menstrual pain were significantly lowered, which proves flattening of curvatures and more vertical insertion of the sacrum. |
|
2. |
The main problem in this study that I concerned is that the authors measured the spinal parameters by using a gravitational inclinometer in a smartphone. This is an indirect method and the results may be influenced by geometry of vertebrae, muscle, and subcutaneous fat tissue. Since the measured topographic points were on the surface of the body, manual palpation of these points may also introduce bias. Traditionally, the spinal parameters should be measured based on the radiography or other medical images. Therefore, the authors should justify the reasonability of the methodology used in this study. |
n/a |
Thank you for your guidance. In the study, the parameters of the spine were measured using an inclinometer in a smartphone, because this method is less invasive than X-ray examination. X-ray radiation is not indifferent to health. The sum of received doses of X-rays over a lifetime can lead to serious illnesses. The use of an invasive test method could lead to a reduction in the number of people participating in the study. Spinal curvature examination using an inclinometer, or a smartphone is a method that provides highly accurate results, comparable to X-ray examination. This is confirmed by numerous studies e.g. 1. Barrett, E.; Lenehan, B.; O’sullivan, K.; Lewis, J.; McCreesh, K. Validation of the manual inclinometer and flexicurve for the measurement of thoracic kyphosis. Physiotherapy Theory and Practice 2017, 34(4), 301–308. doi:10.1080/09593985.2017.1394411. 2. WaÅ›, J.; Sitarski, D.; Ewertowska, P.; Bloda, J.; Czaprowski, D. Using smartphones in the evaluation of spinal curvatures in a sagittal plane. Advances in Rehabilitation 2016, 30(4), 29–38. doi:10.1515/rehab-2015-0053. 3. Saur, P.; Ensink, F.; Frese, K.; Seeger, D.; Hildebrandt, J. Lumbar Range of Motion: Reliability and Validity of the Inclinometer Technique in the Clinical Measurement of Trunk Flexibility. Spine 1996, 21(11), 1332-1338.
|
|
3. |
Another problem is that this study investigated the relationship of spinal alignment and menstrual pain, but not the causal effect of these two factors. Therefore, some representation (e.g., effect on...) in the paper should be modified. |
n/a |
Thank you for your guidance. We have applied the corrections as suggested.
|
|
4. |
Line 91, the reverse of the inclusion criteria should not occur in the exclusion criteria. |
2 |
Thank you for your guidance. We have applied the corrections as suggested. |
|
5. |
Line 121, the authors may add a graph discribing the location of the parameters measured in the study. This could let readers understand the paper more easily. |
3 |
Thank you for your guidance. We have applied the corrections as suggested. We added a graph discribing the location of the parameters measured in the study. |
|
6. |
Some representation can be improved. For example, in "size of curvatures" (line 74), "size of" can be removed. |
2 |
Thank you for your guidance. We have applied the corrections as suggested. |
|
7. |
"Mean body height in CG 165.9 (±6.3) cm" (line 84) lacks predicate. |
2 |
Thank you for your guidance. We have applied the corrections as suggested. |
|
8. |
"Mann-Whitney U test" (line 172). This sentence seems to be imcomplete. |
4 |
Thank you for your guidance. We have applied the corrections as suggested. |
|
9. |
"The results of our study give similar" (line 307). This sentence seems imcomplete. |
9 |
Thank you for your guidance. We have applied the corrections as suggested. |
|
10. |
Therefore, the paper should be polished by a native speaker. |
n/a |
Thank you for your guidance. The article has been reviewed by a native speaker. |

Round 2
Reviewer 3 Report (New Reviewer)
The authors have satisfactorily addressed most of my concerns. I have no further suggestions.
This manuscript is a resubmission of an earlier submission. The following is a list of the peer review reports and author responses from that submission.
Round 1
Reviewer 1 Report
The authors stated that: the aim of presented study was to assess the relationship between the size of curvatures and the mobility of the spine in the sagittal plane on menstrual pain in young women. In my opinion, the topic raised in this study is very important.
The manuscript was prepared with due care and in accordance with the editorial rules required by IJERPH. However, I suggest a few points that need correction. They are listed in detail below.
Introduction:
The content of the "Introduction" allows for a full understanding of the purpose of the presented study. The authors refer to the results of previous research related to the subject of menstrual pain and emphasize that there is a lack of analysis of the relationships between the shape of spinal curvatures and the occurrence of pain during menstruation in women.
Materials and Methods:
- Page 3, lines 97-98: The abbreviation 'the VAS scale' is used. Why don't the authors explain what this abbreviation means. Reference to the publication on the assessment of the effectiveness and safety of exercise for women with primary dysmenorrhea will not release the authors from the obligation to explain the meaning of the abbreviations used in their manuscript.
- Page 4, line 137: The notation of the abbreviation used for middle thoracic kyphosis should be corrected from KKM to TKM.
- Page 4, line 138: The notation of the abbreviation used for upper thoracic kyphosis should be corrected from KKU to TKU.
- Page 4, lines 140-141: Delete 'Dowager's hump/Buffalo hump/' and insert ‘SH’ abbreviation in this sentence. This abbreviation is used in Table 2, Table 4 and Table 5 and is not previously explained.
- Page 4, line 147-148: Separate the word 'performed' with a space from the explanations in parentheses.
- Page 4, line 154: I suggest adding a broader description of ‘the VAS scale’ here. How this scale is built, how it should be used and how the answers are interpreted. In this paragraph, it should be added that the questionnaire also contained questions regarding information on methods of pain relief used by the respondents. The results for this data are discussed on page 6 on lines 215-219, although they are not mentioned in the methods.
- Page 4, lines 157-158: There is no information whether the normality of the distribution of the analysed variables/parameters has been verified. If so, what test was used to verify the distributions of the analysed variables and which variables were normally distributed and which were not. Please complete this information.
- Page 4, lines 158-159: The part of the sentence: ‘which was dictated by the different numbers in the Mann-Whitney U groups’ should be deleted. The explanation that the reason for using the Mann-Whitney U test was the fact that analysed data does not require the equality of the compared groups to be met, should be included in a separate sentence.
Results:
- Page 5, line 172: In the footer below the Table 1 - the explanation 'U – t Mann-Whitney test' should be written as 'the test statistic for the Mann Whitney U Test'. And the explanation that 'p– level of probability' needs to be corrected. It should be written as 'p - probability value'.
- Page 5, lines 176-181: Values of arithmetic means given in the text as a number in degrees should be written with the symbol as (x = 17.1o).
- Page 5, lines 188-190: In the footer below the Table 2 - correct the explanations of the abbreviations BETA, GAMMA, and DELTA in such a way that they are consistent with the information provided in subchapter 2.3.2. 'Assessment of spinal curves'. The symbols used in the table indicate the angles of inclination of specific points on the spine, which should be clearly stated. It can be written, for example, as: ‘BETA - angle of inclination of the Th12-L1 intervertebral space’. For the abbreviations GAMMA and DELTA, respectively.
- Page 5, lines 190-192: In the footer below the Table 2 - correct the explanation of the abbreviations LLA, TKM and TKU. For example, as: 'LLA - lumbar lordosis angle calculated as ALPHA 2 + BETA; TKM - angle of middle thoracic kyphosis calculated as BETA + DELTA; TKU - angle of upper thoracic kyphosis calculated as BETA + GAMMA'.
- Page 6, line 201: In the footer below the Table 3 - the same remark as in the footer to table 1 (Page 5, line 172).
- Page 6, lines 211-214: In the footer below the Table 4 - the same remark as in the footer to table 2 (Page 5, lines 188-190). And the explanation that 'p– level of probability' needs to be corrected. It should be written as 'p - probability value'.
- Page 7, line 227: The reference to Table 5 should be written in parentheses – (Table 5).
- Page 7, Table 5: Why are the abbreviations ALPHA 1, ALPHA 2, BETA, GAMMA, DELTA in lowercase in Table 5, and uppercase in the earlier tables? The same problem occurs in the text before the table on line 226. The notation should be standardized throughout the manuscript.
Discussion:
The discussion is well-written, considering the results of research that addressed problems related to dysmenorrhea and pelvic mobility in women.
Limitation:
- I suggest deleting the sentence: ‘All consenting women from the College of Medical Sciences were tested’. This information is not related to the limitation of the study.
I hope that my comments will allow the authors to improve the manuscript so that it will be of interest to a wide range of scientists and clinical practitioners.

Author Response
Dear Editor-in-Chef,
Please find enclosed the manuscript Effect of lumbar spine mobility and postural alignment on menstrual pain in young women ID: ijerph-2259625, which we would like to submit for publication as a full research paper in: The Role of Physiotherapy and Osteopathy in Gynecology and Obstetrics Issues after changes according to the remarks of reviewers. We thank the reviewers for all their comments. Answers are summarized in the table below. According to the comments, we state that a native speaker has verified the text.

Reviewer 2 Report
The hypothesis raised by the author (changes in spinal mobility and its shape interfere with premenstrual pain) needs to be better supported. Also, why do spinal changes interfere only with premenstrual pain and not at other times?
The methodology is not described in an organized way. For example, the description of the sample casuistry is already made in the methods and not in the results, even before the description of the inclusion criteria. The author describes that he performed a sample calculation but did not specify which parameters he used. It describes ethics committee approval but does not quote any identification number. Author describes that patients were separated into two groups for comparison of characteristics but they were not randomized or included in follow-up groups, therefore it is a single sample divided only for statistical analysis of results. In addition to the comparison, would a correlation test be interesting?
Author Response

(The authors gave the same response as above.)
